# Erythritol Can Inhibit the Expression of Senescence Molecules in Mouse Gingival Tissues and Human Gingival Fibroblasts

**DOI:** 10.3390/nu15184050

**Published:** 2023-09-19

**Authors:** Haruna Yokoi, Masae Furukawa, Jingshu Wang, Yu Aoki, Resmi Raju, Yoriko Ikuyo, Mitsuyoshi Yamada, Yosuke Shikama, Kenji Matsushita

**Affiliations:** 1Department of Oral Disease Research, Geroscience Research Center, National Center for Geriatrics and Gerontology, Obu 474-8511, Japan; haruna.yokoi@ncgg.go.jp (H.Y.); wjsh@ncgg.go.jp (J.W.); karuvachattu@gmail.com (R.R.); yikuyo@ncgg.go.jp (Y.I.); mitsuyos@dpc.agu.ac.jp (M.Y.); shikama@ncgg.go.jp (Y.S.); 2Department of Geriatric Oral Science, Graduate School of Dentistry, Tohoku University, Sendai 980-8575, Japan; 3Research Department, Daiichi Sankyo Healthcare Co., Ltd., Tokyo 140-8710, Japan; aoki.yu.g2@daiichisankyo-hc.co.jp; 4Section of Community Oral Health and Epidemiology, Division of Oral Health, Technology and Epidemiology, Faculty of Dental Science, Kyushu University, Fukuoka 812-8582, Japan; 5Department of Operative Dentistry, School of Dentistry, Aichi Gakuin University, Nagoya 464-8650, Japan

**Keywords:** gingiva, senolytic drugs, cellular senescence, sugar alcohol, oral aging

## Abstract

Oral aging causes conditions including periodontal disease. We investigated how the sugar alcohol erythritol, which has anti-caries effects, impacts aging periodontal tissues and gingival fibroblasts in mice and humans in vivo and in vitro. Mice were classified into three groups: control groups of six-week-old (YC) and eighteen-month-old mice (AC) and a group receiving 5% *w*/*w* erythritol water for 6 months (AE). After rearing, RNA was extracted from the gingiva, and the levels of aging-related molecules were measured using PCR. Immunostaining was performed for the aging markers p21, γH2AX, and NF-κB p65. p16, p21, γH2AX, IL-1β, and TNFα mRNA expression levels were higher in the gingiva of the AC group than in the YC group, while this enhanced expression was significantly suppressed in AE gingiva. NF-κB p65 expression was high in the AC group but was strongly suppressed in the AE group. We induced senescence in cultured human gingival fibroblasts using H_2_O_2_ and lipopolysaccharide before erythritol treatment, which reduced elevated senescence-related marker (p16, p21, SA-β-gal, IL-1β, and TNFα) expression levels. Knockdown of *PFK* or *PGAM* promoted p16 and p21 mRNA expression, but erythritol subsequently rescued pyruvate production. Overall, intraoral erythritol administration may prevent age-related oral mucosal diseases.

## 1. Introduction

The changes in body composition and physiological and metabolic functions seen with aging are related to the aging of the cells and tissues that make up the body [1]. Age-related changes in brain tissue are associated with neurodegenerative diseases, including Alzheimer’s and Parkinson’s diseases [2]. Aging blood vessels and progressive arteriosclerosis trigger hypertension and ischemic diseases [3]. Likewise, the oral cavity exhibits age-related changes. Numerous oral aging-related phenomena exist, such as tooth mobility, wear [4], periodontal disease, and halitosis [5,6,7,8].

Periodontal disease increases in incidence with age, and aging of the periodontal tissue may contribute to this increase. Periodontal tissue supports the teeth and protects them from periodontal pathogens; however, as people age, the structure of periodontal tissue changes, making it more vulnerable to inflammation and impairing wound healing [9].

Aging is characterized by a decline in cellular functions. It delays wound healing by decreasing cell migration, proliferation, and collagen production. In addition, after the cells that constitute the body divide a certain number of times, they exceed the limit of division and irreversibly stop proliferating, a phenomenon known as cellular senescence [10]. During this process, the number of normal cells in an organ decreases and senescent cells develop, resulting in decreased organ weights, atrophy, and impaired physiological function. This increased number of senescent cells may exacerbate chronic inflammation. Senescent cells are characterized by senescence-associated secretory phenotype (SASP) and increased release of inflammation-promoting molecules, including proteases, chemokines, inflammatory cytokines, and chemokines [11]. SASP cells are involved in immune cell recruitment, which affects the oral mucosal microenvironment and influences the function of adjacent cells [12]. Because gingival fibroblasts and epithelial cells acquire this phenotype via the induction of senescence, these senescent cells may promote the occurrence and progression of oral mucosal diseases, including periodontal disease [13,14]. 

Researchers are actively pursuing the development of senolytic drugs, as they may prevent and treat various aging-related diseases by controlling cellular senescence [15]. Quercetin and fisetin have attracted attention as representative low molecular weight senolytic compounds [16]. Both exhibit antioxidant properties and activate sirtuin genes. Quercetin is used in combination with the anticancer drug dasatinib, which is a tyrosine kinase inhibitor. In mouse experiments, it eliminated senescent cells and extended lifespan by 20–30% [17].

However, many of these compounds are not sufficiently effective or safe to be used alone. Hyperglycemia reportedly induces cellular senescence of renal tubular epithelial cells, vascular endothelial cells, and periodontal tissue macrophages [18,19,20] In contrast, sugar alcohols (also called polyols) are hydrogenated carbohydrates that include erythritol, xylitol, and sorbitol. These polyols are often used as sucrose substitutes for their dietary benefits [21]. Xylitol is widely known to prevent dental caries [22]. In addition, erythritol, a sugar alcohol produced from glucose via yeast fermentation, reportedly improves and prevents oral mucosal diseases such as periodontal disease and dental caries [23]. The results of animal and oral caries studies show that sugar alcohols exhibit little to no cariogenicity [24]. Erythritol is reportedly more easily absorbed in the intestine than xylitol, even though they are both sugar alcohols [25], and it has been found to suppress increases in blood glucose levels [26]. However, the effects of erythritol on cellular senescence induced by hyperglycemia and other factors have not yet been clarified.

In this study, we focused on the sugar alcohol erythritol and performed both in vitro and in vivo experiments to investigate the possibility of this compound inhibiting cellular senescence.

## 2. Materials and Methods

### 2.1. Animals

We obtained a six-week-old and eighteen-month-old male C57BL/6N mice (n = 10 per group) from CLEA Japan (Tokyo, Japan). The animals were housed at the National Center for Geriatric Gerontology (NCGG) and Oriental Bio Services (Kobe BM Laboratory, Minami-Yamashiro Laboratory) in climate-controlled clean rooms. All experiments were approved by the Animal Welfare Committee of the NCGG (approval number 4-10-R1) and Oriental Bio Services (approval number 22K010). The experimental habitat had a 12-h light/dark cycle, with lights off from 19:00 to 07:00.

Aged mice were randomly classified into two groups: aged control (AC; normal water, n = 10) and aged erythritol (AE: 5% *w*/*w* erythritol water, n = 10). Meanwhile, the younger mice were placed in the young control (YC; normal water, n = 10) group.

The YC and AC groups were administered normal drinking water, whereas the AE group were administered 5% *w*/*w* erythritol water. All the groups were provided food ad libitum. During the second month of rearing, the mice were kept for 24 h in split rearing (Time HC8 Single; O’Hara and Co., Ltd., Somerdale, NJ, USA) in one cage each (each cage n = 1, total n = 8) to measure daily food and water consumption.

The YC group was euthanized two months after the start of rearing, and the AC and AE groups were euthanized six months after the start of rearing. Mice were anesthetized with an intraperitoneally injected anesthetic (medetomidine hydrochloride 0.9 mg/kg + midazolam 12 mg/kg or alfaxalone 90 mg/kg + butorphanol 15 mg/kg, diluted with saline). They were then humanely euthanized via cervical dislocation and their mandibles, gingivae, and blood were harvested. All experiments were conducted in accordance with the Guide for the Care and Use of Laboratory Animals published by the National Institutes of Health (8th edition). This study followed the Animal Research: Reporting of In Vivo Experiment guidelines.

### 2.2. Body Weight and General Health

Following 7 d of habituation, the mice were weighed with a laboratory scale (±0.01 g) and examined for any indicators of poor health, such as visible injuries and behavioral changes. Body weights were measured weekly.

### 2.3. Blood Tests

#### 2.3.1. Preparation of Blood Sera

Euthanasia was performed between 10:00 a.m. and 12:00 p.m. for blood collection. The mice were decapitated, and 1 mL trunk blood samples were collected in centrifuge tubes (CJ-2AS, Terumo, Tokyo, Japan). Sera were then separated via centrifugation (3000× *g*, 10 min, 15 °C) and stored at −80 °C until the assay. Collected sera were used for biochemical testing.

#### 2.3.2. Serum Biochemistry

Serum biochemical tests (albumin, glucose, and total cholesterol [T-CHO]) were performed at the Nagahama Biological Science Laboratory (Shiga, Japan) of Oriental Yeast Co., Ltd., Tokyo, Japan. At least six mice from each group were analyzed.

### 2.4. Hematoxylin and Eosin Histology

Hematoxylin and eosin (HE) staining was performed at BioPathology Institute. Co., Ltd. (Kunisaki, Oita, Japan). Mandibular gingiva samples were fixed with 10% formalin solution (Fujifilm Wako, Osaka, Japan). After fixation, the tissues were washed under running tap water, dehydrated in ascending grades of ethanol, and cleared in xylene. Paraffin-embedded tissue sections 6 μm in thickness were cut using a microtome and mounted on glass slides. Histology and photomicrography were performed by Biopathology (Kunisaki, Oita, Japan).

### 2.5. Immunofluorescence Analysis

Immunostaining was performed at the Biopathology Institute. Co., Ltd. (Kunisaki, Oita, Japan). After 4% paraformaldehyde fixation, the oral mucosa around the mandibular molars was permeabilized and blocked using Blocking One (03953-95, Nacalai Tesque, Kyoto, Japan). The slides were incubated with antibodies to p21 (ab188224, Abcam, Cambridge, UK) γH2AX (Sigma Aldrich, St. Louis, MO, USA), phospho-histone H2A.X (Ser139) (clone JBW30, 05-636; MerckMillipore, Waltham, MA, USA), and NF-κB p65 (200-301-065, Rockland Immunochemicals, Pottstown, PA, USA) at 4 °C overnight. After incubation, the sections were washed with phosphate buffer saline (PBS).

The Alexa Fluor 488-conjugated secondary antibodies were: chicken anti-rabbit IgG (H + L) (A21441; Invitrogen, Carlsbad, CA, USA), donkey anti-rat IgG (H + L) (A21208; Invitrogen), and donkey anti-mouse IgG (H + L) (A21202; Invitrogen), and sections were counterstained with 4′,6-diamidino-2-phenylindole (DAPI, SJ217; Dojindo, Kumamoto, Japan). We observed the sections with a fluorescence microscope (BZ-X800; Keyence, Osaka, Japan). Images are representative of at least three independent experiments. The numbers of positive cells in the perigingival region were counted using ImageJ v1.52a. Three individuals performed a quantitative analysis of each immunolabeled area, and their scores were averaged.

### 2.6. Cell Culture

Normal human gingival fibroblasts (HGFs, #PCS-201-018; American Type Culture Collection, Manassas, VA, USA) were cultured in high glucose-containing Dulbecco’s modified Eagle’s medium (DMEM; #D5796, Sigma-Aldrich, St. Louis, MO, USA) with 10% FBS or low glucose-containing DMEM (#D6046, Sigma-Aldrich) at 37 °C in a humidified atmosphere with 5% CO_2_. After incubation in 100 mm diameter culture dishes until 70% confluency was reached, the following experiments were performed. 

### 2.7. HGF Senescence Induced by High-Concentration Glucose

We examined gingival fibroblast senescence in vitro. HGFs were cultured in DMEM containing low concentrations of glucose (LG; 6 mM) and passaged in DMEM containing high concentrations of glucose (HG; 25 mM) and in media with 25, 50, and 100 mM erythritol. After 2 d of culture, the cells were analyzed using the following two methods: (1) SA-β-gal staining and (2) real-time PCR experiments, for which total RNA was collected.

### 2.8. Measurement of SA-β-gal

We used a senescence detection kit (BioVision, Waltham, MA, USA) to perform SA-β-gal staining according to the manufacturer’s instructions. The stained cells were counted at 20× magnification in three random fields for each experimental condition [27]. 

### 2.9. Hydrogen Peroxide-Induced HGF Senescence

HGFs were seeded in 100 mm tissue culture dishes at 2 × 10^5^ cells/dish. They were passaged when cultures reached roughly 60–75% confluency for SA-β-gal staining, before 1500 fibroblasts were placed in 35 × 10 mm dishes. N-Acetyl-L-cysteine (NAC) was used to inhibit senescence, following a method adapted from Saxena et al. [28]. NAC (Fujifilm Wako) 2 mM was added, and the culture was pre-dosed for 2 h. After the culture was finished, 300 μM H_2_O_2_ (Fujifilm Wako) was added, and the cells were incubated for 2 h. Then, the medium was changed, and the cells were incubated for 48 h. 

### 2.10. Lipopolysaccharide-Induced HGF Senescence

HGFs were seeded in 100 mm tissue culture dishes at a density of 2 × 10^5^ cells/dish. When cultures reached approximately 60–75% confluency, HGFs were treated with 100 μg/mL lipopolysaccharides from the periodontopathic bacterium *Porphyromonas gingivalis* (LPS-PG; 14946-71, Nacalai Tesque Inc., Kyoto, Japan), incubated for 24 h, and analyzed using two methods: (1) SA-β-gal staining and (2) total RNA collection and real-time PCR for the expression of senescence-related molecules.

### 2.11. Real-Time PCR 

At least six mice per group were used for PCR analysis. We extracted total RNA from the palate gingiva using a NucleoSpin RNA kit (U0955C; Takara Bio, Shiga, Japan) according to the manufacturer’s instructions. We then adjusted the total RNA concentration to 100 ng/μL using a NanoDrop TM2000 spectrophotometer (Thermo Fisher Scientific, Waltham, MA, USA). First-strand cDNA synthesis was performed using a ReverTra Ace-α Kit (Toyobo, Osaka, Japan). FastStart Essential DNA Green Master Mix (Roche, Mannheim, Germany) was used according to the manufacturer’s protocol for PCR amplification, and a LightCycler 96 System (Roche) was used for detection. The primer sequences can be found in Table 1. Target gene expression levels were normalized to that of the housekeeping gene *Gapdh*, and the results were compared with those of the control for each sample. Experiments were conducted in triplicate for each condition. The values are presented as fold-changes between samples using the 2^–ΔΔCt^ method.

### 2.12. Glycolytic Molecule Inhibition Experiment

We knocked down or inhibited phosphofructokinase (*PFK*) and phosphoglycerate mutase (*PGAM1*), which are involved in the glycolytic pathway. PFK was knocked down or suppressed by culturing HGFs in the same manner as described above, using INTERFERin transfection reagent (Polyplus, Illkirch, France) in a six-well plate of cultured human gingival fibroblasts. The final concentration of each siRNA, i.e., *PFK1* siRNA (h) (sc-44561; Santa Cruz Biotechnology [SCBT], Dallas, TX, USA), *PGAM* siRNA (h) (sc-6278; SCBT), was adjusted to 100 nM in accordance with the manufacturer’s protocol, and the three groups were subjected to medium changes 4 h after siRNA addition. For the inhibition experiment, 160 µM of sodium orthovanadate (CAS 13721-39-6; SCBT) was used as a PFK inhibitor, and 56 µM of PGAM1-IN-1 (HY-128681; MedChemExpress, Monmouth, NJ, USA) as a PGAM inhibitor. 

### 2.13. Experiments to Verify the Effect of Erythritol on the Glycolytic System

Experiments were performed to confirm the effects of PFK knockdown (PFKKD) and erythritol on pyruvate, the end products of the glycolytic pathways.

A number of 6-well plates of cultured HGFs were incubated with hPFK siRNA (final concentration: 100 nM) using Polyplus INTERFERin transfection reagent (Polyplus) according to the manufacturer’s protocol, or siRNA + erythritol. After 48 h, the following method was used for analysis: pyruvate was quantified by measuring fluorescence intensity at 530 and 590 nm using a Pyruvate Assay Kit (MET-5029, Cell Biolabs, San Diego, CA, USA).

### 2.14. Statistical Analysis

Values are presented as the mean ± standard error of the mean (SEM). One- and two-way analyses of variance (ANOVA) were used to evaluate the control and experimental groups. If a significant effect was detected, Tukey’s post hoc test was performed, with *p* < 0.05 being considered significant. All statistical analyses were performed using GraphPad Prism 9 Version 9.5.1 (GraphPad Software).

## 3. Results

### 3.1. Changes in Body Weight and Serum Components after Drinking Erythritol Water

To examine the effects of long-term erythritol consumption, 18-month-old male C57BL/6N mice (AC) fed plain water, older C57BL/6N mice (AE) fed 5% erythritol water, and 6-week-old male C57BL/6N mice (YC) fed plain water were kept for 2 or 6 months, and changes in body weight were measured over time (Figure 1a,b). The body weights of the YC group were significantly different from those of the AC and AE groups during the measurement period, but those of the AC and AE groups did not differ significantly from each other (Figure 1b). After 6 months of erythritol water administration, the feed and water intakes of the three groups were compared (Figure 1c,d). There was no difference in daily feed intake between the AC, AE, and YC groups after 6 months of erythritol water administration (YC = 4.08 ± 0.22 g, AC = 3.36 ± 0.4 g, AE = 3.92 ± 0.4 g, YC vs. AC; *p* = 0.32, YC vs. AE; *p* = 0.94, AC vs. AE; *p* = 0.53, Figure 1c). However, the AE group drank significantly more water per day than the AC and YC groups (YC = 6.96 ± 0.5 g, AC = 4.46 ± 0.4 g, AE = 12.23 ± 0.6 g; YC vs. AC; *p* = 0.006, YC vs. AE; *p* < 0.001, AC vs. AE; *p* < 0.001, Figure 1d).

Serum biochemical tests were performed after 6 months of erythritol administration. Albumin (ALB; g/dL) did not differ among the three groups (YC = 3.05 ± 0.08, AC = 3.10 ± 0.03, AE = 3.23 ± 0.06, YC vs. AC; *p* = 0.85, YC vs. AE; *p* = 0.22, AC vs. AE; *p* = 0.47, Figure 1E), T-CHO levels were similar among the three groups (YC= 86.6 ± 7.05, AC= 87.3 ± 8.85, AE = 88.0 ± 7.51, YC vs. AC; *p* = 0.998, YC vs. AE; *p* = 0.992, AC vs. AE group; *p* = 0.998, Figure 1f). Serum glucose levels did not differ among the three groups (YC = 200 ± 12.84, AC = 204 ± 12.49, AE = 217.5 ± 18.89, YC vs. AC; *p* = 0.98, YC vs. AE; *p* = 0.71, AC vs. AE; *p* = 0.82, Figure 1g). Administration of erythritol water did not affect the nutritional status of aged mice.

### 3.2. Effect of Erythritol Administration on Aging of Gingiva in Aged Mice

In addition, evolution of the gingival sulcus was observed in the inner margin epithelium, along with the disappearance of the adherent epithelium. In contrast, in the AE group, the thickening of the epithelium did not improve, but the structural changes in the basal layer observed in the AC group improved, and the loss of adherent epithelium was suppressed (Figure 2a).

Next, the levels of p21 and γH2AX proteins, which are indicators of cellular senescence, were analyzed in the same tissues using immunostaining. p21 and γH2AX proteins were hardly produced in the gingival tissues of the YC group, but their production was significantly higher in AC group gingiva (p21: YC vs. AC, *p* = 0.002; YC vs. AE, *p* = 0.3; AC vs. AE, *p* = 0.02. γH2AX: YC vs. AC, *p* < 0.001; YC vs. AE, *p* = 0.25; AC vs. AE, *p* = 0.02. Figure 2b,c). p21 was found in the gingival epithelium, and γH2AX was mainly enhanced in the submucosal tissues. In contrast, the number of cells positive for p21, γH2AX protein, and their cognate molecules was significantly less increased in the gingiva of the AC group than in the AE group. In addition, the mRNA expression of p16 was significantly increased along with that of p21 in the gingiva of YC mice, whereas its expression was significantly decreased in the gingiva of the AE group (p16: YC vs. AC. *p* < 0.001; YC vs. AE, *p* < 0.001; AC vs. AE, *p* < 0.001; p21: YC vs. AC, *p* < 0.001; YC vs. AE, *p* < 0.01; AC vs. AE, *p* = 0.12; Figure 2d). Next, the mRNA expression levels of the SASP factors TNFα and IL-1β in the gingiva were compared among the three groups. The results showed that IL-1β and TNFα mRNA expression was significantly increased in the gingiva of AC compared to YC (IL-1β: YC vs. AC, *p* < 0.001; YC vs. AE, *p* < 0.05; AC vs. AE, *p* < 0.001. TNFα: YC vs. AC, *p* < 0.001; YC vs. AE, *p* = 0.17; AC vs. AE, *p* < 0.05; Figure 2d). However, the expression levels were significantly suppressed in the AE group treated with 5% erythritol (Figure 2d). We examined the expression of NF-κB p65, which is associated with the expression of many SASP factors, including TNFα and IL-1β. We found that its expression was significantly higher at the protein and positive cell levels in the gingival tissues of the AC group compared to those in the YC group. However, these values were significantly lower in the gingiva of the AE group (YC vs. AC, *p* < 0.001; YC vs. AE, *p* = 0.13; AC vs. AE, *p* < 0.01, Figure 2d,c).

### 3.3. Anti-Aging Effect of Erythritol on Cultured Gingival Fibroblasts

In the HG group, about 22% of cells were SA-β-gal-positive after 48 h of culture. However, the addition of erythritol decreased the expression of SA-β-gal-positive cells in a volume-dependent manner (Figure 3a,b). The mRNA expression of the senescence-associated molecules p16 and p21 was also significantly reduced by the addition of 50 and 100 mM erythritol (HG vs. HG + E50, *p* < 0.001; HG vs. HG + E100, *p* < 0.001).

Next, we examined the effect of erythritol on cultured human gingival fibroblasts after senescence, which was induced via the addition of hydrogen peroxide or LPS from the periodontopathic bacterium *Porphyromonas gingivalis*. Specifically, 300 µM hydrogen peroxide or 100 µg/mL LPS was added to human gingival fibroblast culture systems and incubated for 24 h. Next, 100 mM erythritol was added, and aging-related molecule expression was analyzed. The positive control used was 2 mM NAC. More SA-β-gal positive cells (about 39%) were observed in the H_2_O_2_-treated (H) group than in the C group, but the addition of NAC (N + H) or erythritol (E + H) significantly reduced the number of SA-β-gal positive cells (Figure 4a,b). The expression of senescence-related molecules p16, p21, IL-1β, and TNFα was also strongly induced by H_2_O_2_, but their expression was significantly suppressed by the addition of erythritol along with NAC (Figure 4c–f). Positive-stained cells (approximately 70%) were observed in the LPS-treated group (Figure 5a,b). However, the addition of NAC or erythritol significantly suppressed the increase in SA-β-gal positive cells. (Figure 5a,b) mRNA expression levels of senescence-related molecules p16, p21 IL-1β, and TNFα were also enhanced via LPS addition, but their expression was significantly suppressed in the NAC or erythritol-treated groups (Figure 5c–f).

### 3.4. Relationship between the Glycolytic System, Cellular Senescence, and the Effects of Erythritol

To explore the mechanism by which erythritol inhibits cellular senescence, we examined its effect on the glycolytic system. We focused on glycolytic metabolic enzymes, phosphofructokinase (PFK), or phosphoglycerate mutase (PGAM) and inhibited their mRNA expression in cultured human gingival fibroblasts using siRNA. Cultured human gingival fibroblasts in which PFK was knocked down showed significantly more SA-β-gal positive cells. However, the addition of erythritol to PFK-knockdown HGFs significantly decreased SA-β-gal-positive cells (Figure 6a). This was also observed following PGAM1-knockdown (PGMAKD) of HGFs. However, the addition of erythritol to the same cells decreased the number of SA-β-gal-positive cells (Figure 6a). p16 mRNA expression was increased with the knockdown of PFK and PGAM1, but their expression was significantly suppressed with the addition of erythritol (Figure 6b). In addition, PFK knockdown significantly enhanced p21 mRNA expression, which was suppressed via the addition of erythritol. However, its expression was not altered in PGAM1-knockdown cells and was not affected by the addition of erythritol (Figure 6c).

Next, we examined the effect of erythritol on HGFs when small-molecular-weight compounds that inhibit the enzymatic activities of PFK and PGAM were added. The increased number of SA-β-gal-positive cells was significantly suppressed in the erythritol + PFK inhibitor group (PFKnib group). Likewise, a significant increase in the number of SA-β-gal-positive cells was observed in the PGAM inhibitor group (PGAMnib group), which was reduced with the addition of erythritol (Figure 6b). p16mRNA expression was enhanced in the PFKnib and PGAMnib groups but was significantly suppressed with the addition of erythritol (Figure 6e). Furthermore, p21mRNA expression was enhanced in the PFKnib group but was significantly reduced with the addition of erythritol (Figure 6f). However, no enhancement in p21mRNA expression was observed in the PGAMnib group, and its expression was unchanged upon the addition of erythritol (Figure 6d–f). We further examined the effect of erythritol on the production of pyruvate, an intermediate metabolite of the glycolytic system. The amount of pyruvate produced in the cells also increased in the PFKKD group, whereas a significant increase in pyruvate production was observed in the PFKKD + E group (Figure 7a). Furthermore, the amount of pyruvate also increased in the PGAMKD group, but not significantly; an increase in the amount of pyruvate was observed in the PGAMKD + E group (Figure 7b). These results suggest that erythritol inhibits cellular senescence by modulating intracellular glycolytic metabolism and enhancing pyruvate production.

## 4. Discussion

In this study, we examined the effects of erythritol on the expression of senescence- and inflammation-related molecules in vivo and in vitro using mouse gingival tissue and human gingival fibroblast culture systems, respectively. The expression of senescence-associated molecules and inflammatory cytokine mRNAs were greater in the gingiva older mice compared to younger mice, and these expression levels were suppressed when aged mice were treated with erythritol, reducing them to a level similar to that found in young mice. In cultured human gingival fibroblasts, erythritol inhibited the induction of cellular senescence to the same extent as NAC, a known senescence inhibitor. Furthermore, erythritol may inhibit cellular senescence by modulating the intracellular metabolism of the lytic cycle, ultimately enhancing pyruvate production. These results suggest that erythritol may be a promising new senolytic drug.

Cellular senescence is associated with the development of atherosclerosis, dementia, and cancer [29]. Further, mouse models have shown that eliminating senescent cells can inhibit the development of these diseases [30,31]. Various reagents that selectively kill senescent cells (senolytic drugs) have been reported, and their efficacy has been demonstrated in animal or clinical studies [32,33,34]. Dasatinib and quercetin are typical examples of such drugs [35]. Dasatinib, an anticancer drug, inhibits multiple tyrosine kinases and induces the apoptosis of senescent cells. Quercetin, a flavonoid, also inhibits many kinases and serpins, including PI3K, and promotes the apoptosis of senescent cells. Simultaneous administration of dasatinib and quercetin to spontaneously aged mice inhibited senescence and prolonged their lifespan [36]. However, safety concerns have been reported for some senolytic drugs, and they have not been demonstrated to be safe for long-term administration [37].

Erythritol is a naturally occurring tetracarbohydrate sugar alcohol found in many fruits and fermented foods [38], which is approximately 75% as sweet as sucrose. It is a “glucose fermentation sweetener” produced from glucose via yeast fermentation and is the only sugar with zero calories. Against the backdrop of increasing health consciousness, erythritol is widely used as a sweetening ingredient in low-calorie products and sugarless confections [39]. Excessive intake of sugar alcohols is known to cause nausea, bloating, diarrhea, and other unpleasant gastrointestinal issues. The low absorption of sugar alcohols, which cause osmotic effects and intestinal water retention, has been blamed for this [40]. Further, unabsorbed polyols may be fermented by the intestinal microflora, producing gas. However, erythritol is mostly absorbed in the small intestine, with only a small amount reaching the colon. Erythritol is generally better tolerated and exerts fewer gastrointestinal side effects than comparable amounts of other polyols. There is extensive evidence from both human and animal studies on the absorption, metabolism, and excretion of erythritol, as well as short- and long-term toxicity studies examining possible reproductive, developmental, genotoxic, mutagenic, and carcinogenic effects, making it a safe compound for participants. Thus, erythritol-containing products can be considered safe for living organisms [23,39]. Erythritol is an existing food product, and no serious side effects have been observed with its long-term use. Therefore, it is a promising candidate senolytic drug with a good safety profile.

Cellular senescence and the glycolytic system are closely related. Overexpression of PGAM in mouse embryonic fibroblasts (MEFs) enhances the entire glycolytic system, and the cells remain proliferative without senescence [41]. In contrast, when PGAM1 mRNA expression was decreased in MEFs by siRNA, early cell senescence was induced [42]. In addition, p53, one of the markers of cellular senescence, directly regulates the expression of several glycolytic genes (*GLUT1*, *GLUT4*, *HK2*) and also suppresses PFK via TIGAR (TP53-induced glycolysis and apoptosis regulator), a downstream factor of p53 that suppresses PFK [43]. In addition, P53 suppresses PGAM via Mdm2 [42]. Thus, cellular senescence-related factors are targets of glycolytic metabolism, and the regulation of the glycolytic system may be an important factor in inducing cellular senescence; therefore, cellular senescence may be inhibited via regulation of the glycolytic system. 

We found that the expression of the senescence-associated molecules SA-βgal, p16, and p21, which are increased via knockdown of *PFK* and *PGAM1* mRNA or inhibition of their enzymatic activities, was restored to control levels by erythritol (Figure 6). Intracellular pyruvate production was enhanced with the addition of erythritol (Figure 7). These results suggested that erythritol suppresses cellular senescence by enhancing glycolytic metabolism. However, the molecular mechanisms by which erythritol regulates glycolysis remain unknown. A sub-pathway of glucose metabolism (the polyol circuit) exists, in which the molecule is metabolized to the sugar alcohol sorbitol with aldose reductase, and to fructose with sorbitol dehydrogenase [44]. However, no metabolic pathway that uses erythritol as a substrate is known. We analyzed the molecular mechanisms by which erythritol regulates the glycolytic system, including the existence of such pathways. While this investigation yielded some clues, one limitation of our study is that the molecular mechanisms by which erythritol regulates the glycolytic system in oral cells remain unclear. Further investigations to address this issue are warranted.

The various benefits of erythritol have been reported in the dental–oral field. Experiments using rats have reported that erythritol is not utilized by *Streptococcus mutans*, which are caries-causing bacteria in the oral cavity, and it does not cause dental caries itself [45]. In vitro experiments have shown that, like xylitol, erythritol inhibits the growth of *S. mutans* and the formation of biofilms [46]. Because oral bacteria cannot metabolize erythritol, it does not cause dental caries [23]. Similar to xylitol, it has antibacterial activity against streptococci and may prevent dental caries by reducing plaque formation [23]. Further, it also exerts antibacterial effects against periodontal bacteria [47,48]. Periodontitis occurrence increases with age, and aging is considered a risk factor for periodontitis [49]. In addition, an association between periodontal tissue aging and the pathogenesis of periodontal diseases has been suggested [9,13,14]. The results of the present study revealed that erythritol inhibits gingival tissue and gingival fibroblast senescence, making it an innovative compound that ameliorates not only the causative bacteria, but also the fragility of host tissues. We expect that dental pastes and rinses containing erythritol will be developed and used widely in the future.

## 5. Conclusions

Erythritol ameliorated age-related histopathological changes in gingival tissue and inhibited gingival tissue and fibroblast senescence in a mouse model and a cell culture system. Erythritol regulates cellular senescence by modulating the glycolytic system. Future applications of erythritol in the oral cavity are expected to lead to the development of new oral care methods targeting the inhibition of oral tissue aging.

## Figures and Tables

**Figure 1 nutrients-15-04050-f001:**
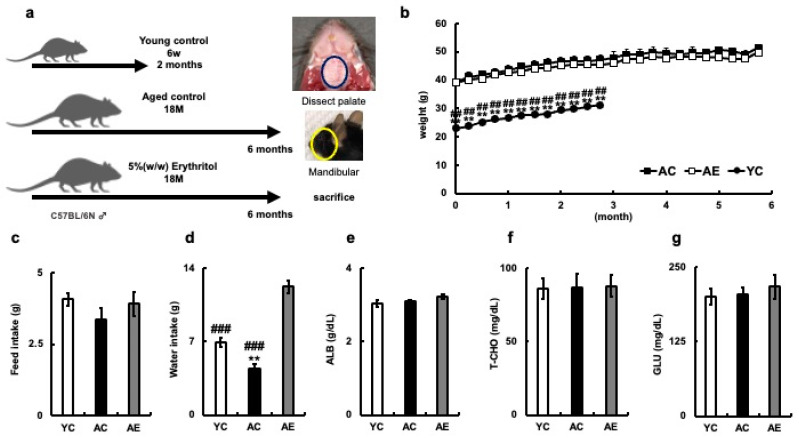
Systemic effects of erythritol in mice. (**a**) Experimental schedule. YC mice were euthanized 2 months after rearing; AC and AE mice were euthanized 6 months after rearing. (**b**) Percent body weight gain of each mouse. (**c**) Daily food intake of mice (**g**). (**d**) Daily water consumption of mice (**g**); (**e**) Albumin (ALB) of mouse serum (g/dL); (**f**) Total cholesterol (T-CHO; mg/dL); (**g**) Glucose (GLU; mg/dL). YC, young control; AC, aged control; AE, aged erythritol. One-way ANOVA, Tukey’s post hoc test. ** *p* < 0.01, ## *p* < 0.01, ### *p* < 0.001. * indicates YC vs. AC, # indicates YC vs. AE, AC vs. AE. Results are presented as mean ± SE; SE, standard error; ANOVA, analysis of variance.

**Figure 2 nutrients-15-04050-f002:**
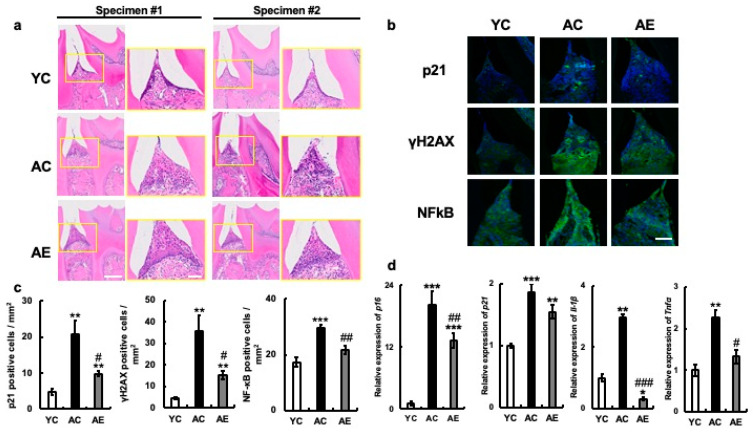
Effects of erythritol on mouse oral tissue. (**a**) H&E staining of periodontal suite in mouse mandibular tissue. H&E staining of mouse mandibular periodontal tissue. Specimens 1 and 2 are different individuals. Enlarged yellow squares in the figure on the left are shown on the right, respectively. Scale bar indicates 50 μm and 100 μm. (**b**) Periodontal tissue immunostaining in mouse mandibular p21, γH2AX, and NF-κB p65, respectively. Scale bar indicates 100 μm. (**c**) Number of cells positive for p21, γH2AX, and NF-κB in periodontal tissues of mouse mandible. (**d**) Aging-related mRNA expression in mouse maxillary gingiva: YC, young control; AC, aged control; AE, aged erythritol. One-way analysis of variance and Tukey’s post hoc test. * *p* < 0.05, ** *p* < 0.01, *** *p* < 0.001, # *p* < 0.05, ## *p* < 0.01, ### *p* < 0.001. * indicates YC vs. AC and YC vs. AE, # indicates AC vs. AE. Results are presented as mean ± SE; SE, standard error; ANOVA, analysis of variance.

**Figure 3 nutrients-15-04050-f003:**
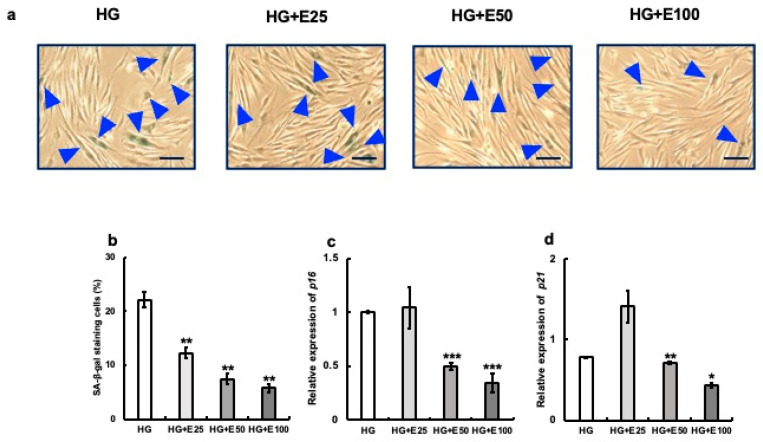
Effect of erythritol on human gingival fibroblasts (HGFs) cultured in high-concentration glucose. HGFs cultured under the following conditions were stained with SA-β-gal: HG (HGFs cultured in high glucose concentration), HG + E25 (HGFs cultured in high-concentration glucose + 25 mM erythritol), HG + E50 (HGFs cultured in high-concentration glucose + 50 mM erythritol), and HG + E100 (HGFs cultured in high-concentration glucose + 100 mM erythritol); blue triangles indicate SA-β-gal-positive cells. Scale bar indicates 100 μm. (**a**) The SA-β-gal-positive cells in (**a**) are shown in the graph. (**b**) The SA-β-gal-positive cells shown in (**b**) are also shown in the graph. (**c**,**d**) p16 and p21 mRNA expression in cells under each condition. One-way analysis of variance and Tukey’s post hoc test. * *p* < 0.05, ** *p* < 0.01, *** *p* < 0.001, * denotes HG vs. HG+; SE, standard error; ANOVA, analysis of variance.

**Figure 4 nutrients-15-04050-f004:**
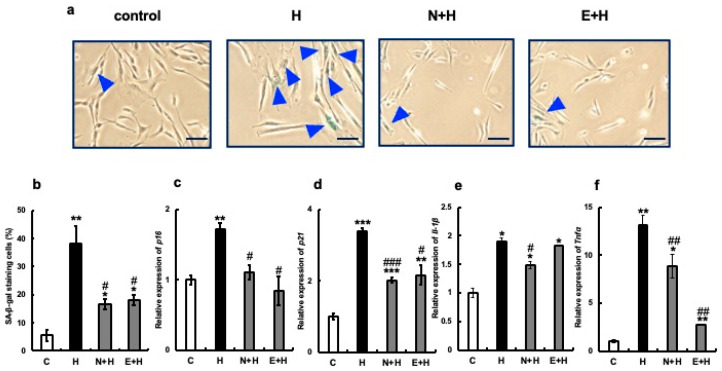
Effect of erythritol on the induction of senescence by hydrogen peroxide in human gingival fibroblasts (HGFs). (**a**) Human gingival fibroblasts (HGFs) were cultured under the following conditions and stained for SA-β-gal: C (control HGFs cultured in regular DMEM), H_2_O_2_ (HGFs cultured in medium with a high concentration of H_2_O_2_), N + H (HGFs cultured in medium with NAC pretreatment + high concentration of H_2_O_2_), E + H (HGFs cultured in medium with a high concentration of H_2_O_2_) HGF + E + H (HGFs + erythritol cultured in medium with a high concentration of H_2_O_2_), blue triangles indicate SA-β-gal positive cells. Scale bar indicates 100 µm. (**b**) SA-β-gal positive cells shown in (**a**) are shown in the graph. (**c**–**f**) Expression of p16, p21, IL-1β, and TNFα. One-way analysis of variance and Tukey’s post hoc test. * *p* < 0.05, ** *p* < 0.01, *** *p* < 0.001, # *p* < 0.05, ## *p* < 0.01, ### *p* < 0.001. * indicates C vs. H, N + H, E + H comparison; # H vs. N + H, E + H comparison. Results are presented as mean ± SE; SE, standard error; SE, standard error; ANOVA, analysis of variance.

**Figure 5 nutrients-15-04050-f005:**
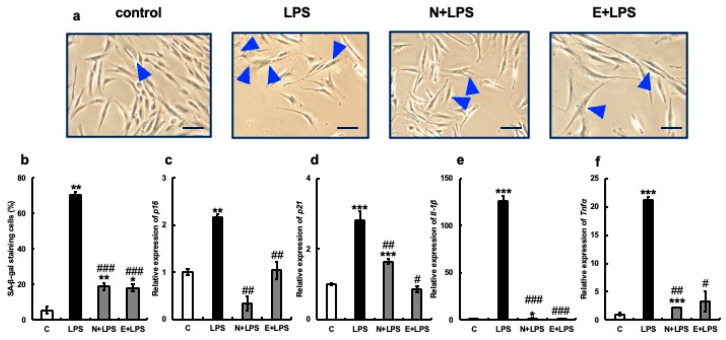
Effect of erythritol on LPS (lipopolysaccharide)-induced senescence of human gingival fibroblasts (HGFs). (**a**) HGFs were cultured and SA-β-gal stained under the following conditions: C (control HGFs cultured in normal DMEM), LPS (HGFs cultured in LPS-added medium), N + LPS (HGFs cultured in NAC pretreatment + LPS-added medium), E + LPS (HGFs cultured in LPS + erythritol-added medium); blue triangles indicate SA-β-gal positive cells. Scale bar indicates 100 µm. (**b**) Graph showing SA-β-gal-positive cells. (**c**–**f**) p16, p21, IL-1b, and TNFα mRNA expression is shown. One-way ANOVA and Tukey’s post hoc test. * *p* < 0.05, ** *p* < 0.01, *** *p* < 0.001, # *p* < 0.05, ## *p* < 0.01, ### *p* < 0.001. * C vs. L, N + L, and E + L comparisons; # L vs. N + L and E + L comparisons. Results are presented as mean ± SE. SE, standard error; ANOVA, analysis of variance.

**Figure 6 nutrients-15-04050-f006:**
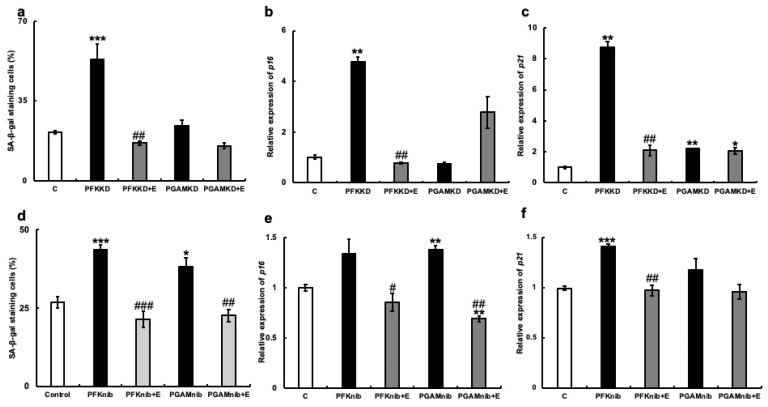
Effect of erythritol under the suppression of glycolytic molecules in human gingival fibroblasts (HGFs). HGFs were cultured under the following conditions and examined for SA-β-gal staining and mRNA expression of p16 and p21: C (HGFs cultured in regular DMEM), PFKKD (HGFs cultured with PFK siRNA), PFKKD + E (PFKKD with erythritol), PGAMKD (HGFs cultured with PGAM siRNA). (**a**) SA-β-gal positive cells shown. (**b**,**c**) mRNA expression of p16 and p21 was examined. (**d**) SA-β-gal positive cells shown. (**e**,**f**) p16 and p21 mRNA expression levels. One-way ANOVA and Tukey’s post hoc test. * *p* < 0.05, ** *p* < 0.01, *** *p* < 0.001, # *p* < 0.05, ## *p* < 0.01, ### *p* < 0.001. * C vs. PFKKD, KD + E, nib, nib+E; # corresponds to comparison of KD vs. KD + E and nib vs. to nib+E. Results are presented as mean ± SE. SE, standard error; ANOVA, analysis of variance.

**Figure 7 nutrients-15-04050-f007:**
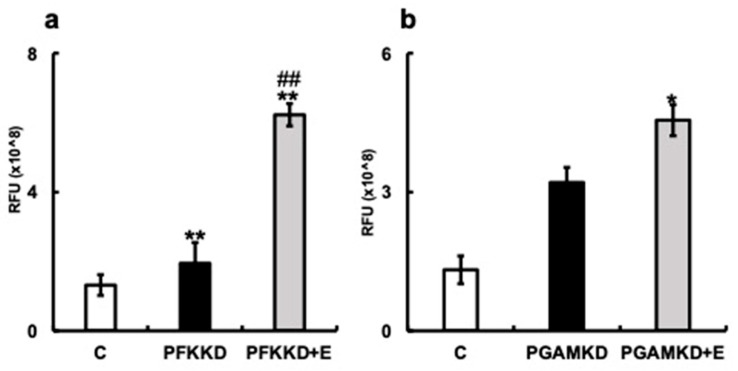
Effect of erythritol under inhibition of glycolytic molecules in human gingival fibroblasts (HGFs). HGFs were cultured under the following conditions and RFUs (relative fluorescent unit) were measured: C (HGFs cultured in normal DMEM), PFKKD (HGF PFKKD with siRNA), PFKKD + E (HGF PFKKD cultured in medium with siRNA and erythritol) (**a**,**b**) Pyruvate levels. * *p* < 0.05, ** *p* < 0.01, ## *p* < 0.01. C vs. * indicates PFKKD, C vs. PFKD + E, C vs. PGAMKD, and C vs. PGAMKD + E comparisons. # corresponds to comparison of PFKKD vs. PFKD + E, PGAMKD vs. PGAMKD + E. Results are presented as mean ± SE; SE, standard error; ANOVA, analysis of variance.

**Table 1 nutrients-15-04050-t001:** Sequences of primers used for real-time PCR.

Name	Forward	Reverse
Mouse *p16*	CGTACCCCGATTCAGGTGAT	TTGAGCAGAAGAGCTGCTACG
Mouse *p21*	GTGGGTCTGACTCCAGCCC	CCTTCTCGTGAGACGCTTAC
Mouse *Il-1β*	GCACTACAGGCTCCGAGATGAAC	TTGTCGTTGCTTGGTTCTCCTTGT
Mouse *Tnfα*	ATGAGCACAGAAAGCATGA	AGTAGACAGAAGAGCGTGGT
Mouse *Gapdh*	AACCTGCCAAGTATGATGA	GGAGTTGCTGTTGAAGTC
Human *p16*	CTCGTGCTGATGCTACTGAGGA	GGTCGGCGCAGTTGGGCTCC
Human *p21*	CCGAAGTCAGTTCCTTGTGG	CATGGGTTCTGACGGACAT
Human *IL-1β*	GCAGCCATGGCAGAAGTACCTGA	CCAGAGGGCAGAGGTCCAGGTC
Human *TNFα*	AGGCGCTCCCCAAGAAGACA	TCCTTGGCAAAACTGCACCT
Human *GAPDH*	TGTCAGTGGTGGACCTGACCT	AGGGGAGATTCAGTGTGGTG

*p16*, cyclin-dependent kinase inhibitor 2A; *p21*, cyclin-dependent kinase inhibitor 1; *IL-1β*, interleukin 1 beta; *TNFα*, tumor necrosis factor alpha; *GAPDH,* Glyceraldehyde-3-phosphate dehydrogenase.

## Data Availability

The data presented in this paper are available on request from the corresponding author.

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
