# Peer review of "Erythritol Can Inhibit the Expression of Senescence Molecules in Mouse Gingival Tissues and Human Gingival Fibroblasts"

_nutrients, 2023, doi:10.3390/nu15184050_

Round 1
Reviewer 1 Report
Dear authors,
The manuscript entitled "Erythritol can inhibit the expression of senescence molecules in mice gingival tissues and human gingival fibroblasts" is very interesting and relevant.
This manuscript presents, in a well-structured manner, the inhibitory effect of erythritol on the senescence molecules of mice gingival tissues and human gingival fibroblasts.
The experiments were approved by the 92 Animal Welfare Committee of the NCGG (approval number 4-10-R1) and Oriental Bio 93 Services (approval number 22K010).
The authors presented the aims of their research: they investigated the possibility of the sugar alcohol erythritol inhibiting cellular senescence using in vitro and in vivo experimental systems.
The manuscript is scientifically sound.
Materials and methods are clearly presented. All these experiments were realized in accordance with the Animal Research: Reporting of In Vivo Experiments guideline. There were presented the mouse groups; the euthanization method; the harvest of studied tissues (mandibles, gingivae) and of blood; the weekly determination of wounds and body weights; the blood tests; real-time PCR; Histopathological study with hematoxylin-eosin staining; Immunofluorescence analysis; Cell culture; HGF senescence induced by high-concentration glucose Measurement of SA--gal; Hydrogen peroxide-induced HGF senescence lipopolysaccharide-induced HGF senescence's inhibition experiment of glycolytic molecules; Experiments to verify the effect of erythritol on the pentose phosphate pathway or glycolytic system The statistical analysis was performed using GraphPad Prism 8 (GraphPad Software).
The results presented were the correlation between the changes in body weight and serum components after drinking erythritol water; the effect of erythritol administration on the aging of gingiva in aged mice; the anti-aging effect of erythritol on cultured gingival fibroblasts; the relationship between the glycolytic system, cellular senescence, and the effects of erythritol. The presented figures (7 in number) are clear and appropriate. The results are reproducible based on the details given in the methods section.
The discussion section also provides the results of other authors on the studied problem, but most of the citations are more than 5 years old (30, 33, 36, 39, 42-46, 48, 49, 51, 54).
The conclusions are clearly presented and argued, underlying the idea that intraoral erythritol administration may prevent age-related oral mucosal diseases. The cited references are relevant, but of a total of 54, 27 references are older than 5 years (the bibliographic titles 4-6, 8-10, 12, 19, 21-23, 25, 27, 28, 30, 33, 36, 39, 42-46, 48, 49, 51, 54).
For the reasons given above, I consider that the manuscript should make corrections relating to the cited references.
Reviewer 2 Report
1. The results of YC (young control) were not shown in Figure 1.
2. Is it possible to add more groups of the HE results in Figure 2?
3. Is it possible to detect the SA-β-gal-positive cells with clear and same white background in Figure 3, 4 and 5?
4. The identification of glycolytic molecules involved in this process in Figure 6 and 7 was a little too simple.
5. The correlation among senescence, inflammation and glycolysis were not discussed completely.
1. The English language could be reviewed by a professional editor.
2 Some small questions: Capital letter was used in figure legends, while Lowercase letter was used in pictures. The "t" and "p" used to show significant difference values should be in italics.
Round 2
Reviewer 1 Report
Dear Authors,
Congratulations!
It is a hard-working study with adequate detail, which is why the manuscript fully deserves to be published.